# MANAGING SOLUTION STABILITY IN DECISION-FOCUSED LEARNING WITH COST REGULARIZATION

## ABSTRACT

Decision-focused learning is an emerging paradigm that integrates predictive modeling and combinatorial optimization by training models to directly improve decision quality rather than prediction accuracy alone. Differentiating through combinatorial optimization problems represents a central challenge, and recent approaches tackle this difficulty by introducing perturbation-based approximations that enable end-to-end training. In this work, we focus on estimating the objective function coefficients of a combinatorial optimization problem. We analyze how the effectiveness of perturbation-based techniques depends on the intensity of the perturbations, by establishing a theoretical link to the notion of solution stability in combinatorial optimization. Our study demonstrates that fluctuations in perturbation intensity and solution stability can lead to ineffective training. We propose addressing this issue by introducing a regularization of the estimated cost vectors which improves the robustness and reliability of the learning process. Extensive experiments on established benchmarks show that this regularization consistently improves performance, confirming its practical benefit and general applicability.

## 1 INTRODUCTION

Recent advances in data processing and analytics have led to a shift in operations management for supply chains, healthcare systems, and industries (Mišić & Perakis, 2020). Data is integrated into decision-making processes by training Machine Learning (ML) models to create mappings between observed features and the parameters of an optimization problem (Qi & Shen, 2022). In most situations, prediction errors from ML models result in suboptimal decisions and unnecessary costs that could have been prevented (Cameron et al., 2021). A growing body of research is advancing the *Decision-Focused Learning* (DFL) paradigm, which tackles this challenge by embedding combinatorial optimization into the learning process to enhance prediction quality for specific optimization tasks.

We focus in this work on estimating costs in the objective function of combinatorial optimization problems taking the form of *Mixed-Integer Linear Programs* (MILP). These problems are modeled by continuous and discrete variables, with a linear objective function to optimize and a set of linear constraints to respect. For this specific task, decision-focused learning most often outperforms standard ML models, but it is hindered by the piecewise constant nature of decision models mappings. Nevertheless, a large variety of approximate techniques has been proposed in recent years to circumvent this difficulty and learn from discrete decision models, most of them relying on decisions perturbations to identify relevant descent directions. In practice, managing the intensity of these perturbations can be challenging and may result in underwhelming performance.

Our primary contribution is to expose a limitation shared by all examined perturbation-based DFL techniques, grounded in key principles of optimal solution stability in combinatorial optimization. Specifically, we demonstrate that, depending on the perturbation intensity relative to the estimated costs, certain methods may resort to imitating known solutions rather than improving a target performance metric, while others may become entirely ineffective. Whereas previous work often overlooks perturbation intensity—leading to degraded learning behavior—our study shows that regularizing the estimated costs provides better control over the decision-focused learning process, a finding supported by extensive numerical experiments on multiple established benchmarks.

## 2 RELATED WORK

The integration of prediction and optimization in ML models has gained significant attention in recent years for its potential to improve decision-making under uncertainty (Kotary et al., 2020; Mandi et al., 2024). Mandi et al. (2024) outline several categories of DFL techniques dedicated to the prediction of an optimization problem objective. A first category analytically differentiates certain optimization mappings via their Lagrangian formulations (Amos & Kolter, 2017). While powerful in theory, this approach requires additional smoothing for linear programs and the relaxation of integrality constraints for mixed-integer problems, introducing substantial approximations when applied to MILPs (Wilder et al., 2019). Because our focus is on perturbation-based methods for MILP optimization, we do not further consider this line of work.

Another category relies on constructing smooth approximations of optimization mappings by the introduction of random perturbations. Originating from implicit differentiation via parameter perturbations (Domke, 2010), this idea was adapted for DFL by considering distributions over decisions rather than single optimal solutions, enabling differentiation through discrete mappings. This adaptation produced the *Perturb-and-MAP* framework (MAP) (Papandreou & Yuille, 2011), later refined in Implicit Maximum Likelihood Estimation (IMLE) (Niepert et al., 2021), which improved perturbation sampling. The *Differentiable Black-Box* (DBB) approach (Pogancic et al., 2021) built on similar principles but employed different perturbation strategies, while *Differentiable Perturbed Optimizers* (DPO) (Berthet et al., 2020) extended Perturb-and-MAP by treating the optimization mapping as a deterministic function of both cost estimates and perturbations.

The last category gathers techniques with a dedicated loss function rather than a general differentiation framework. We take an interest in two state-of-the-art loss functions among the existing approaches. First, the *Smart Predict-the-Optimize* (SPO) loss (Elmachtoub & Grigas, 2022) corresponds to a surrogate regret loss function whose sub-gradients indicates worthwhile descent directions. We show that it relies on the introduction of a perturbation for optimization mapping differentiation, hence the need for regularization. Secondly, the *Fenchel-Young* loss (Blondel et al., 2020) is a differentiable imitation loss function designed to reproduce the ground truth optimal decision. We compare the previously mentioned approaches to this loss function to analyze when learning behaviors shift from performance-driven improvement to simple imitation.

From an optimization perspective, Bengio et al. (2021) introduce a distinction broadly categorizing DFL techniques as learning either by imitation of expert behavior, or by experience aimed at improving performance. We adopt this classification to demonstrate that perturbation-based DFL techniques designed for learning by experience may become inefficient—or collapse into imitation—if the cost vector estimates are not properly regularized. To support this claim, we draw on classical results related to solution stability in combinatorial optimization (Bonnans & Shapiro, 2000).

While some studies mention regularization of cost vector estimates, it is typically introduced as a problem-specific adjustment—for instance, to address particular DFL applications (Rolinek et al., 2020) or as part of an extension of the DBB approach (Sahoo et al., 2022). Blondel et al. (2020) also mention regularization, although it is of a different nature as it applies to the optimization mapping for differentiation rather than to the cost vector itself. To date, regularization has not been recognized as a broadly applicable practice, independent of the chosen DFL method or the underlying optimization problem.

## 3 GENERAL FRAMEWORK

Consider a convex cone $\Theta \subset \mathbb{R}^n$ and a non-empty finite set of distinct points $\mathbb{Y} \subset \mathbb{R}^n$ with $\mathcal{C}$ its convex hull. We consider a general MILP with linear costs parameterized by a vector $\boldsymbol{\theta} \in \Theta$ and an associated *optimization mapping* $f : \Theta \mapsto \mathcal{C}$ defined as follows:

$$\min_{y \in \mathcal{C}} \boldsymbol{\theta}^{\mathsf{T}} y, \quad f(\boldsymbol{\theta}) = \arg\min_{y \in \mathcal{C}} \boldsymbol{\theta}^{\mathsf{T}} y \tag{1}$$

Function $f$ maps a cost vector $\boldsymbol{\theta} \in \Theta$ to an optimal decision $f(\boldsymbol{\theta}) \in \mathcal{C}$. Our goal is to estimate a cost vector $\boldsymbol{\theta}$ whose quality depends on the resulting decision $f(\boldsymbol{\theta})$.

## 3.1 Learning from Decisions

We design a machine-learning model described by equations:

$$\boldsymbol{\theta} = h_{\boldsymbol{v}}(\boldsymbol{x}), \quad \boldsymbol{y} = f(\boldsymbol{\theta}) \tag{2}$$

Parameter $\boldsymbol{x} \in \mathbb{X}$ denote feature inputs while $\boldsymbol{y} \in \mathbb{Y}$ is a target output representing a solution to a discrete decision model parameterized by $\boldsymbol{\theta} \in \Theta$. Function $h_{\boldsymbol{v}} : \mathbb{X} \mapsto \Theta$ is a differentiable parameterized mapping while $f : \Theta \mapsto \mathbb{Y}$ represents an optimization mapping as described above. Here, the optimization mapping can be seen as a combinatorial optimization layer within our ML model: it can be incorporated at any layer of our decision-focused learning pipeline. Moreover, we consider a training loss function defined as:

$$L(\boldsymbol{x}, \bar{\boldsymbol{\theta}}; \boldsymbol{v}) = l(\boldsymbol{y}, \bar{\boldsymbol{\theta}}), \quad \text{with } \boldsymbol{y} = f(h_{\boldsymbol{v}}(\boldsymbol{x})) \tag{3}$$

This DFL approach aims to predict the cost objective associated with the optimization mapping. Given a dataset $\{(\boldsymbol{x}_j, \bar{\boldsymbol{\theta}}_j)\}_{j=1}^N$, we learn parameters $\boldsymbol{v}$ of equation 2 to minimize $L(\boldsymbol{x}, \bar{\boldsymbol{\theta}}; \boldsymbol{v})$. In other words, we aim at learning a model that minimizes the evaluation value of a decision $\boldsymbol{y} = f(\boldsymbol{\theta})$ made based on the cost vector estimate $\boldsymbol{\theta} = h_{\boldsymbol{v}}(\boldsymbol{x})$. This evaluation value may assess the quality of the decision estimate $f(\boldsymbol{\theta})$ according to the ground truth cost vector $\bar{\boldsymbol{\theta}}$ and the ground truth decision $f(\bar{\boldsymbol{\theta}})$.

This is typically observed when training a machine learning model using the *regret* loss $L^r$, which quantifies how much worse the decision based on predicted parameters is compared to the decision based on the true parameters. Specifically, it evaluates the decision estimate $f(\boldsymbol{\theta}) \in \mathcal{C} \subset \mathbb{R}^n$ with respect to the ground-truth cost vector $\bar{\boldsymbol{\theta}} \in \Theta \subset \mathbb{R}^n$.

$$L^r(\boldsymbol{x}, \bar{\boldsymbol{\theta}}; \boldsymbol{v}) = \bar{\boldsymbol{\theta}}^\mathsf{T} \boldsymbol{y} - \bar{\boldsymbol{\theta}}^\mathsf{T} f(\bar{\boldsymbol{\theta}}), \quad \text{with } \boldsymbol{y} = f(h_{\boldsymbol{v}}(\boldsymbol{x})) \tag{4}$$

Although our examples and numerical experiments focus on this specific loss function and model structure, our contributions generalize to all studied perturbation-based DFL techniques within the broader framework introduced in equation 2.

Differentiating any training loss function $L$ as defined in equation 3 with regards to $\boldsymbol{v}$ is necessary in order to train our model. According to equation 3, this gradient can be expressed as:

$$\nabla_{\boldsymbol{v}} L(\boldsymbol{x}, \bar{\boldsymbol{\theta}}; \boldsymbol{v}) = \nabla_{\boldsymbol{v}} h_{\boldsymbol{v}}(\boldsymbol{x})^\mathsf{T} \nabla_{\boldsymbol{\theta}} L(\boldsymbol{x}, \bar{\boldsymbol{\theta}}; \boldsymbol{v}) \tag{5}$$

Hence it is necessary to differentiate the training loss function with regards to $\boldsymbol{\theta}$. By chain rule this particular gradient can be expressed as follows:

$$\nabla_{\boldsymbol{\theta}} L(\boldsymbol{x}, \bar{\boldsymbol{\theta}}; \boldsymbol{v}) = \nabla_{\boldsymbol{y}} L(\boldsymbol{x}, \bar{\boldsymbol{\theta}}; \boldsymbol{v})^\mathsf{T} \nabla_{\boldsymbol{\theta}} f(\boldsymbol{\theta}) \tag{6}$$

We conclude that in the case of the regret loss $L^r$, one has:

$$\nabla_{\boldsymbol{y}} L^r(\boldsymbol{x}, \bar{\boldsymbol{\theta}}; \boldsymbol{v}) = \bar{\boldsymbol{\theta}} \tag{7}$$

$$\nabla_{\boldsymbol{\theta}} L^r(\boldsymbol{x}, \bar{\boldsymbol{\theta}}; \boldsymbol{v}) = \bar{\boldsymbol{\theta}}^\mathsf{T} \nabla_{\boldsymbol{\theta}} f(\boldsymbol{\theta}) \tag{8}$$

Differentiating this loss function in a decision-focused learning model is challenging mainly because $\nabla_{\boldsymbol{\theta}} L^r(\boldsymbol{x}, \bar{\boldsymbol{\theta}}; \boldsymbol{v})$ depends on $\nabla_{\boldsymbol{\theta}} f(\boldsymbol{\theta})$. The optimization mapping $f$ is typically assumed to be piecewise constant, so the gradient $\nabla_{\boldsymbol{\theta}} f(\boldsymbol{\theta})$ is zero almost everywhere. Since this gradient has limited relevance in an optimization context, DFL techniques are employed to identify meaningful descent directions for minimizing the loss function.

## 3.2 Stability Properties of an Optimization Mapping

It is commonly assumed that the optimization mapping $f$ is piecewise constant and thus possesses a gradient that is of little relevance in an optimization process (Mandi et al., 2024). We strengthen this claim by reframing it into a stronger formalism based on combinatorial optimization literature related to optimal solution stability (Bonnans & Shapiro, 2000). Moreover, we highlight that solution stability deeply affects the learning process of perturbation-based DFL techniques. The presented results are illustrated by an example in annex A.

Let us denote by $F$ a set valued optimization mapping such that $F(\boldsymbol{\theta})$ represents the exhaustive set of optimal solutions for the problem defined by equation 1 with cost vector $\boldsymbol{\theta} \in \Theta$. In other words, $F(\boldsymbol{\theta})$ represents the set of all possible values taken by $f(\boldsymbol{\theta})$. We study the concept of upper semi-continuity for the set valued optimization mapping $F$ as presented by Bonnans & Shapiro (2000).

**Definition 1** *The set valued optimization mapping $F$ is upper semi-continuous at $\boldsymbol{\theta}$ if for every open set $\mathcal{U}$ containing $F(\boldsymbol{\theta})$, there exists a neighborhood $\mathcal{V}$ of $\boldsymbol{\theta}$ such that for all $\tilde{\boldsymbol{\theta}} \in \mathcal{V}$, one has $F(\tilde{\boldsymbol{\theta}}) \subset \mathcal{U}$.*

Bonnans & Shapiro (2000) identify sufficient conditions for a set valued mapping to be upper semi-continuous. These conditions are met in our case, given that polytope $\mathcal{C}$ is non-empty and bounded as the convex hull of the non-empty finite set $\mathbb{Y}$. Hence the following property:

**Property 1** *Let us consider the set valued optimization mapping $F$ for the problem defined by equation 1. The mapping $F$ is upper semi-continuous everywhere in $\Theta$.*

In most cases, the optimal solution set $F(\boldsymbol{\theta})$ consists only of a singleton $f(\boldsymbol{\theta})$ (Elmachtoub & Grigas, 2022). We restrict our study to that general case in what follows. In that case, there exists a largest positive scalar $\eta$ such that every perturbed cost vector $\tilde{\boldsymbol{\theta}} \in \Theta$ lying within distance $\eta$ of $\boldsymbol{\theta}$ shares the same optimal decision set as $\boldsymbol{\theta}$, that is, $F(\tilde{\boldsymbol{\theta}}) = F(\boldsymbol{\theta})$. Borrowing from the nomenclature related to stability analysis, the maximum distance $\eta$ is defined as the *stability radius* of cost vector $\boldsymbol{\theta}$, and the neighborhood it defines around $\boldsymbol{\theta}$ is its *stability region* (Sotskov et al., 1995). We conclude that, in most cases, the set valued optimization mapping $F$ is constant within a stability region around $\boldsymbol{\theta}$, depending on a stability radius $\eta$. Hence the following property holds.

**Property 2** *Consider an optimization mapping $f : \Theta \mapsto \mathcal{C}$ and a cost vector $\boldsymbol{\theta}$ such that its optimal decision set $F(\boldsymbol{\theta})$ is a singleton $f(\boldsymbol{\theta})$. Let us further consider a perturbation $\boldsymbol{\delta} \in \Theta$ of the cost vector $\boldsymbol{\theta}$ such that its optimal decision set $F(\boldsymbol{\delta})$ is a singleton $f(\boldsymbol{\delta})$.*

- *If $\boldsymbol{\theta} + \boldsymbol{\delta}$ is in the stability region of $\boldsymbol{\theta}$ then $f(\boldsymbol{\theta} + \boldsymbol{\delta}) = f(\boldsymbol{\theta})$*

- *If $\boldsymbol{\theta} + \boldsymbol{\delta}$ is in the stability region of $\boldsymbol{\delta}$ then $f(\boldsymbol{\theta} + \boldsymbol{\delta}) = f(\boldsymbol{\delta})$*

Example: given the optimization mapping $f : \mathbb{R}^2 \mapsto \mathbb{R}^2$ in Annex A, let us consider a cost vector $\boldsymbol{\theta} \in \mathbb{R}^2$ such that $\boldsymbol{\theta}_1 < \boldsymbol{\theta}_2$. Its stability region is $\{\tilde{\boldsymbol{\theta}} \in \mathbb{R}^2; \tilde{\boldsymbol{\theta}}_1 < \tilde{\boldsymbol{\theta}}_2\}$ since all cost vectors with the same order of costs share the same optimal decision $(1, 0)$. Hence for any perturbation $\boldsymbol{\delta}$ such that $\boldsymbol{\theta} + \boldsymbol{\delta}$ is in the stability region of $\boldsymbol{\theta}$, i.e. $\boldsymbol{\theta}_1 + \boldsymbol{\delta}_1 < \boldsymbol{\theta}_2 + \boldsymbol{\delta}_2$, the optimal decision $f(\boldsymbol{\theta} + \boldsymbol{\delta})$ is also equal to $f(\boldsymbol{\theta}) = (1, 0)$.

This stability property in face of additive perturbation implies that the optimization mapping at $\boldsymbol{\theta} \in \Theta$ remains constant for any perturbation $\boldsymbol{\delta}$ within its stability region. This supports the claim that the optimization mapping is generally piecewise constant which explains the little relevance of its gradient in an optimization process. Moreover, property 2 represents the main motivation to an efficient management of solution stability in decision-focused learning. We define the *scale* of a vector to refer to any scalar factor (including but not limited to its norm) by which the vector may be multiplied. This abstraction allows us to analyze the directional components independently of scale. We now present another essential property of optimization mappings: their scale invariance.

**Property 3** *Let us consider an optimization mapping $f : \Theta \mapsto \mathcal{C}$. For all cost vectors $\boldsymbol{\theta} \in \Theta$ and positive scalar $\alpha > 0$, one has $f(\alpha \cdot \boldsymbol{\theta}) = f(\boldsymbol{\theta})$.*

Example: given the optimization mapping $f : \mathbb{R}^2 \mapsto \mathbb{R}^2$ in Annex A, let us consider a cost vector $\boldsymbol{\theta} \in \mathbb{R}^2$ such that $\boldsymbol{\theta}_1 < \boldsymbol{\theta}_2$. All cost vectors with the same order of costs share the same optimal decision $(1, 0)$. Hence for any scalar $\alpha > 0$, one has $f(\alpha \cdot \boldsymbol{\theta}) = f(\boldsymbol{\theta})$ since one has $\alpha \cdot \boldsymbol{\theta}_1 < \alpha \cdot \boldsymbol{\theta}_2$.

Consider a cost vector $\boldsymbol{\theta} \in \Theta$ of stability radius $\eta$. Then according to property 3, for any perturbation $\boldsymbol{\delta} \in \Theta$ and a positive scalar $\alpha$, one has: $f(\alpha \cdot \boldsymbol{\theta} + \boldsymbol{\delta}) = f(\boldsymbol{\theta} + \alpha^{-1} \cdot \boldsymbol{\delta})$. Hence if $\alpha^{-1} \cdot \boldsymbol{\delta}$ is within distance $\eta$ of $\boldsymbol{\theta}$, then $\boldsymbol{\delta}$ is in the stability region of $\alpha \cdot \boldsymbol{\theta}$. We conclude that the stability radius of $\alpha \cdot \boldsymbol{\theta}$ is $\alpha \cdot \eta$. Hence, a cost vector has its stability radius proportional to its scale and the following property holds.

**Property 4** *Consider two cost vectors $\boldsymbol{\theta}$, $\boldsymbol{\delta} \in \Theta$. The decision $f(\boldsymbol{\theta} + \boldsymbol{\delta})$ depends on a comparison of scale between $\boldsymbol{\theta}$ and $\boldsymbol{\delta}$. If $\boldsymbol{\theta}$ is of arbitrarily greater scale than $\boldsymbol{\delta}$ then one has $f(\boldsymbol{\theta} + \boldsymbol{\delta}) = f(\boldsymbol{\theta})$. Conversely if $\boldsymbol{\theta}$ is of arbitrarily lower scale than $\boldsymbol{\delta}$ then one has $f(\boldsymbol{\theta} + \boldsymbol{\delta}) = f(\boldsymbol{\delta})$.*

Example: given the optimization mapping $f : \mathbb{R}^2 \mapsto \mathbb{R}^2$ in Annex A, let us consider two cost vectors $\boldsymbol{\theta}, \boldsymbol{\delta} \in \mathbb{R}^2$ such that $\boldsymbol{\theta}_1 < \boldsymbol{\theta}_2$ and $\delta_1 > \delta_2$. Remember that all cost vectors with the same order of costs share the same optimal decision in that example. Hence if $\boldsymbol{\theta}$ is of arbitrarily greater scale than $\boldsymbol{\delta}$ then one has $||\boldsymbol{\delta}|| < \boldsymbol{\theta}_2 - \boldsymbol{\theta}_1$ and $\boldsymbol{\theta}_1 + \delta_1 < \boldsymbol{\theta}_2 + \delta_2$: we conclude that $f(\boldsymbol{\theta} + \boldsymbol{\delta}) = f(\boldsymbol{\theta})$.

Now, we show how solution stability affects the learning process of perturbation-based DFL techniques. These techniques rely on a perturbed decision mapping $\tilde{f} : \Theta \mapsto \mathcal{C}$ that introduces an additive perturbation to the input cost vector before evaluation of the corresponding optimization mapping. In other word for a cost vector $\boldsymbol{\theta} \in \Theta$ and a cost perturbation $\boldsymbol{\delta} \in \Theta$, the corresponding perturbed decision is:

$$\tilde{f}(\boldsymbol{\theta}) = f(\boldsymbol{\theta} + \boldsymbol{\delta}) \tag{9}$$

Hence perturbation-based DFL techniques rely on an additive cost perturbation to identify perturbed decisions in the neighborhood of $f(\boldsymbol{\theta})$. These perturbed decisions are then compared to $f(\boldsymbol{\theta})$ in order to identify a descent direction improving the performances with regards to the training loss. Hence, it is necessary that they are close to but different from $f(\boldsymbol{\theta})$.

According to property 4, for an arbitrary lower scale of perturbation $\boldsymbol{\delta}$ compared to that of $\boldsymbol{\theta}$, all perturbed decisions $f(\boldsymbol{\theta} + \boldsymbol{\delta})$ are actually equal to the exact decision $f(\boldsymbol{\theta})$. In this situation, the neighborhood around $f(\boldsymbol{\theta})$ is not effectively explored, making it impossible to identify a descent direction and thus rendering differentiation hardly feasible. Conversely if the perturbation $\boldsymbol{\delta}$ is much larger in scale than $\boldsymbol{\theta}$, the perturbed decision is equal to $f(\boldsymbol{\delta})$. Because $f(\boldsymbol{\delta})$ generally lies far from the neighborhood of $f(\boldsymbol{\theta})$, the resulting descent direction is misleading, and the differentiation process fails to behave as intended. We conclude that the perturbation scale must be chosen proportional to the scale of the cost vector estimate for the differentiation process to be effective.

We have outlined intuitions on how solution stability can negatively affect DFL learning processes. In particular, we emphasized the critical role of comparing the scales of perturbations and cost vector estimates in DFL methods that use additive perturbations for differentiation. In the following sections, we present a detailed analysis of these DFL techniques to gain a deeper understanding of the implications of this phenomenon.

## 4 IMPACTS OF SOLUTION STABILITY ON THE LEARNING PROCESS

We propose in this section an in-depth study of state-of-the-art DFL techniques with regard to the consequences of solution stability on their respective differentiation approach. A distinction is first established between two learning paradigms: imitation and experience. We then show that state-of-the-art DFL techniques that are expected to learn by experience may see their performances degraded by the effect of solution stability, leading either to non-differentiability or to a paradigm shift toward imitation.

### 4.1 LEARNING FROM DECISIONS BY IMITATION OR EXPERIENCE

Bengio et al. (2021) classify DFL techniques based on whether they learn by imitation or by experience. In the imitation setting, optimal decisions are inferred by replicating expert-provided behavior, without direct optimization of a performance measure. In contrast, experience-based learning involves iterative trial-and-error, guided by rewards and penalties, with the explicit goal of optimizing a defined performance metric. Our objective is to demonstrate that, without controlling solution stability, DFL techniques designed for learning through experience become either inefficient or collapse into imitation-based learning. We regard this outcome as equally undesirable, since broadly applicable methods—intended to work with any loss function—would be reduced to merely replicating existing solutions, regardless of the performance they are meant to enhance.

We first identify a characteristic sufficient to classify a DFL technique as learning by imitation. To this end, we examine the Fenchel–Young (FY) loss, which is widely regarded as a standard imitation loss function (Dalle et al., 2022). The FY loss is introduced by Blondel et al. (2020) and corresponds to the cost comparison of the ground-truth decision $f(\bar{\boldsymbol{\theta}})$ and a perturbed estimate decision $\tilde{f}(\boldsymbol{\theta})$. It can be differentiated with regards to $\boldsymbol{\theta}$ such that:

$$\nabla_{\boldsymbol{\theta}} L^{FY}(\boldsymbol{\theta}, \bar{\boldsymbol{\theta}}) = f(\bar{\boldsymbol{\theta}}) - \tilde{f}(\boldsymbol{\theta}) \tag{10}$$

Gradient $\nabla_{\boldsymbol{\theta}} L^{FY}(\boldsymbol{\theta}, \bar{\boldsymbol{\theta}})$ indicates a descent direction toward a local minimum. Following this direction produces cost vector estimates whose optimization mapping best replicates the ground-truth decisions. Hence, the FY loss is clearly associated with a learning process by imitation.

In the following we show that under certain conditions, the gradient estimates of several state-of-the-art DFL techniques can resemble that of the Fenchel–Young loss, effectively categorizing them as learning processes by imitation.

### 4.2 Smart Predict-then-Optimize

Let us examine the Smart-Predict-then-Optimize (SPO) loss introduced by Elmachtoub & Grigas (2022). The SPO loss is presented as a differentiable surrogate for the original regret loss defined in equation 4. Hence it should be expected that it corresponds to a straight-forward learning process by experience dependent on the regret performance measure. This surrogate loss $L_\alpha^{SPO}$ is defined according to a positive scalar $\alpha$, and Elmachtoub & Grigas (2022) determine that it admits a sub-gradient with regards to the $\boldsymbol{\theta}$ such that:

$$\nabla_{\boldsymbol{\theta}} L_\alpha^{SPO}(\boldsymbol{\theta}, \bar{\boldsymbol{\theta}}) \approx f(\bar{\boldsymbol{\theta}}) - f(\alpha\boldsymbol{\theta} - \bar{\boldsymbol{\theta}}) \tag{11}$$

For an arbitrary lower scale of $\alpha\boldsymbol{\theta}$ compared to that of $\bar{\boldsymbol{\theta}}$, one has $f(\alpha\boldsymbol{\theta} - \bar{\boldsymbol{\theta}}) = f(-\bar{\boldsymbol{\theta}})$ according to property 4. Hence the gradient estimate of the SPO loss takes the form of the following difference in decisions:

$$\nabla_{\boldsymbol{\theta}} L_\alpha^{SPO} \approx f(\bar{\boldsymbol{\theta}}) - f(-\bar{\boldsymbol{\theta}}) \tag{12}$$

In this case, all information related to the cost vector estimate $\boldsymbol{\theta}$ is lost in the computation of the gradient estimate. In other words, this gradient estimate cannot indicate a descent direction starting from $\boldsymbol{\theta}$ that would improve the training loss. Hence, differentiation is not efficient.

Conversely if the scale of $\alpha\boldsymbol{\theta}$ is arbitrarily greater than that of $\bar{\boldsymbol{\theta}}$, then $f(\alpha\boldsymbol{\theta} - \bar{\boldsymbol{\theta}}) = f(\boldsymbol{\theta})$ according to property 4. Hence the gradient estimate of the SPO loss takes the form of the following difference in decisions:

$$\nabla_{\boldsymbol{\theta}} L_\alpha^{SPO} \approx f(\bar{\boldsymbol{\theta}}) - f(\boldsymbol{\theta}) \approx \nabla_{\boldsymbol{\theta}} L^{FY}(\boldsymbol{\theta}, \bar{\boldsymbol{\theta}}) \tag{13}$$

In this scenario, the SPO loss gradient aligns with the Fenchel–Young loss gradient, causing the learning process to degenerate from experience-based learning to imitation-based learning.

We conclude that there could be a shift in learning behavior when training a model with the SPO loss function, depending on the stability of solution $f(\alpha\boldsymbol{\theta} - \bar{\boldsymbol{\theta}})$ and on the respective scales of $\alpha\boldsymbol{\theta}$ and $\bar{\boldsymbol{\theta}}$.

### 4.3 Implicit Differentiation by Perturbations

In this section, we investigate a category of DFL techniques that construct smooth approximations of the optimization mappings by adopting a probabilistic point of view. This approach covers three DFL techniques : Perturb-and-MAP introduced by Papandreou & Yuille (2011), Implicit Maximum Likelihood Estimator introduced by Niepert et al. (2021) and Differentiable BlackBox introduced by Pogancic et al. (2021). These methods are considered to be "blackbox" since they depend neither on the optimization problem nor on the considered loss function. Hence we assume they are part of an ML model that learn from decisions by experience.

These methods map the cost vector $\boldsymbol{\theta}$ to a probability distribution $p(\boldsymbol{y}|\boldsymbol{\theta})$ over the solution set $\mathcal{C}$. For a given cost vector $\boldsymbol{\theta}$, an approximation of the exact optimal decision $f(\boldsymbol{\theta})$ can be obtained from the expectation of decisions according to the probability distribution $p(\boldsymbol{y}|\boldsymbol{\theta})$, that is denoted $\mathbb{E}_p[\boldsymbol{y}|\boldsymbol{\theta}] \approx f(\boldsymbol{\theta})$. The perturbed decision $\mathbb{E}_p[\boldsymbol{y}|\boldsymbol{\theta}]$ can be seen as the expected value vector of a distribution of decisions centered at $f(\boldsymbol{\theta})$.

Implicit differentiation by perturbation is originally introduced by Domke (2010). Its application in decision-focused learning leads to the following results. Given a positive scalar $\beta > 0$, the training loss function can see its gradient with regards to $\boldsymbol{\theta}$ be approximated such that:

$$\nabla_{\boldsymbol{\theta}} L(\boldsymbol{x}, \bar{\boldsymbol{\theta}}; \boldsymbol{v}) \approx \mathbb{E}_p[\boldsymbol{y}|\boldsymbol{\theta} + \beta \cdot \nabla_{\boldsymbol{y}} L(\boldsymbol{x}, \bar{\boldsymbol{\theta}}; \boldsymbol{v})] - \mathbb{E}_p[\boldsymbol{y}|\boldsymbol{\theta}] \tag{14}$$

All three previously mentioned approaches rely on this approximation for decision mapping differentiation. For all three methods, the distribution $p(\boldsymbol{y}|\boldsymbol{\theta})$ is obtained by introduction of an additive perturbation to the cost vector. In practice, a finite number of sample are drawn from a

random multivariate variable $\mathbf{d}$ of values in $\Theta$ to approximate the expectation of decisions such that $\mathbb{E}_p[\boldsymbol{y}|\boldsymbol{\theta}] = \mathbb{E}_{\mathbf{d}}[f(\boldsymbol{\theta} + \mathbf{d})]$. If the scale of all samples is arbitrarily greater than that of $\boldsymbol{\theta}$, then one has $\mathbb{E}_p[\boldsymbol{y}|\boldsymbol{\theta}]$ being approximated by $\mathbb{E}_{\mathbf{d}}[f(\mathbf{d})]$. In other words, if the perturbation is of greater scale than cost vector $\boldsymbol{\theta}$, then we no longer study decision in the neighborhood of $\boldsymbol{\theta}$ and all information relating to $\boldsymbol{\theta}$ is lost. We conclude that this differentiation approach is no longer efficient if the scale of perturbation is higher than that of the cost vector estimate.

We now study the difference of scale between cost vectors $\boldsymbol{\theta}$ and $\beta \cdot \nabla_{\boldsymbol{y}} L(\boldsymbol{x}, \bar{\boldsymbol{\theta}}; \boldsymbol{v})$. If the scale of $\boldsymbol{\theta}$ is arbitrarily greater, then according to property 4 one has :

$$\nabla_{\boldsymbol{\theta}} L(\boldsymbol{x}, \bar{\boldsymbol{\theta}}; \boldsymbol{v}) \approx \mathbb{E}_p[\boldsymbol{y}|\boldsymbol{\theta}] - \mathbb{E}_p[\boldsymbol{y}|\boldsymbol{\theta}] = 0 \tag{15}$$

Hence differentiation is hardly feasible since the neighborhood around $f(\boldsymbol{\theta})$ remains unexplored and no descent direction can be found. On the contrary, if the scale of $\boldsymbol{\theta}$ is arbitrarily lower than that of $\beta \cdot \nabla_{\boldsymbol{y}} L(\boldsymbol{x}, \bar{\boldsymbol{\theta}}; \boldsymbol{v})$, then according to property 4 one has:

$$\nabla_{\boldsymbol{\theta}} L(\boldsymbol{x}, \bar{\boldsymbol{\theta}}; \boldsymbol{v}) \approx \mathbb{E}_p[\boldsymbol{y}|\nabla_{\boldsymbol{y}} L(\boldsymbol{x}, \bar{\boldsymbol{\theta}}; \boldsymbol{v})] - \mathbb{E}_p[\boldsymbol{y}|\boldsymbol{\theta}] \tag{16}$$

Similarly to the Fenchel-Young loss, following this gradient estimate produces cost vector estimates whose optimization mapping best replicates the ground-truth decisions. Hence in this case, this approach to differentiation would be similar to that of a learning process by imitation.

For example, when studying the regret loss $L^r$ defined in equation 4, according to equation 7 one would have:

$$\nabla_{\boldsymbol{\theta}} L^r(\boldsymbol{x}, \bar{\boldsymbol{\theta}}; \boldsymbol{v}) \approx \mathbb{E}_p[\boldsymbol{y}|\bar{\boldsymbol{\theta}}] - \mathbb{E}_p[\boldsymbol{y}|\boldsymbol{\theta}] \approx \nabla_{\boldsymbol{\theta}} L^{FY}(\boldsymbol{\theta}, \bar{\boldsymbol{\theta}}) \tag{17}$$

In this case, the regret loss gradient corresponds to the FY loss gradient. Hence we observe that a scale of $\boldsymbol{\theta}$ arbitrarily lower than that of $\beta \cdot \bar{\boldsymbol{\theta}}$ causes the learning process to degenerate from experience-based learning to imitation-based learning.

To conclude, it is essential for all introduced perturbations to be at scale with the cost vector estimate so that DFL techniques relying on implicit differentiation by perturbation remain efficient.

## 4.4 DIFFERENTIABLE PERTURBED OPTIMIZERS

In this section, we investigate Differentiable Perturbed Optimizers (Berthet et al., 2020). This technique is similar to implicit differentiation by perturbations since it is a blackbox approach that relies on the perturbation of the optimal decision estimate in order to explore its neighborhood and identify direction descent of improvement. Therefore, we assume it is part of a ML model that learns from decisions by experience.

The DPO approach considers a probability distribution on the perturbed cost vector $\boldsymbol{\theta}$ rather than on the optimal decision $f(\boldsymbol{\theta})$. A perturbed optimizer $f_\epsilon(\cdot)$ is introduced as a deterministic function of the cost vector $\boldsymbol{\theta}$ and a multivariate random variable $\mathbf{d}$. This random variable acts as an additive perturbation on the cost vector with a temperature $\epsilon$, such that $f_\epsilon(\boldsymbol{\theta}) = \mathbb{E}[f(\boldsymbol{\theta} + \epsilon \cdot \mathbf{d})]$. Assuming that the random variable $\mathbf{d}$ has a density proportional to $\exp(-\nu(\mathbf{d}))$ with function $\nu$ being twice-differentiable, the perturbed optimizer $f_\epsilon(\cdot)$ has its derivative taking the form:

$$\nabla_{\boldsymbol{\theta}} f_\epsilon(\boldsymbol{\theta}) = \mathbb{E}[f(\boldsymbol{\theta} + \epsilon \cdot \mathbf{d}) \cdot \nabla_{\mathbf{d}} \nu(\mathbf{d})^\mathsf{T}] \tag{18}$$

This gradient is in turn integrated into the gradient estimation of the training loss according to equation 5.

In practice, the random variable $\mathbf{d}$ follows either a Gaussian or Gumbel distribution law for which one has $\mathbb{E}[\nabla_{\mathbf{d}} \nu(\mathbf{d})] = 0$. In practice, gradient $\nabla_{\boldsymbol{\theta}} f_\epsilon(\boldsymbol{\theta})$ is estimated by means of a Monte-Carlo sampling over the random variable $\mathbf{d}$. If for all samples, cost vector $\boldsymbol{\theta}$ is of arbitrarily greater scale than $\epsilon \cdot \mathbf{d}$ then according to property 4 one has $f(\boldsymbol{\theta} + \epsilon \cdot \mathbf{d})$ being approximated by $f(\boldsymbol{\theta})$ and the gradient estimate is of zero value. In this case, the perturbation is too low to identify suitable descent decisions in the neighborhood of $f(\boldsymbol{\theta})$, hence differentiation is hardly feasible.

On the contrary, if $\boldsymbol{\theta}$ is of arbitrarily lower scale than all samples then differentiation is possible, but some information is lost. Indeed, according to property 4, it follows that one has $f(\boldsymbol{\theta} + \epsilon \cdot \mathbf{d})$ being approximated by $f(\mathbf{d})$ and:

$$\nabla_{\boldsymbol{\theta}} f_\epsilon(\boldsymbol{\theta}) = \mathbb{E}[f(\mathbf{d}) \cdot \nabla_{\mathbf{d}} \nu(\mathbf{d})^\mathsf{T}] \tag{19}$$

In other words, the perturbation is too great and the gradient estimate for the DPO approach loses all information on the cost vector estimate $\boldsymbol{\theta}$. Hence it is not able to indicate a descent direction starting from $\boldsymbol{\theta}$ that would improve the performances evaluated by the training loss.

We conclude that, for effective differentiation, the perturbation introduced in the DPO approach must be on the same scale as the cost vector estimate.

## 5 MANAGING SOLUTION STABILITY WITH COST REGULARIZATION

We have shown that the general learning behavior emerging from perturbation-based DFL techniques is deeply affected by the comparison in scale between the estimated cost vector and the additive perturbation. In this section, we aim to examine the practical causes of this issue within the training process and explore ways to address it through cost regularization.

In practice, the scale of perturbation depends on the chosen parameters value of each DFL techniques during the learning phase. It is therefore obvious that setting hyperparameters on an appropriate scale is necessary to ensure that gradient estimates remain meaningful throughout the training phase. However, this alone does not prevent the adverse effects of perturbation mismanagement during learning. Indeed, hyperparameters value remains constant throughout the training phase, while the weights of the ML model and its outputs evolve over the process, potentially altering their scale. On one hand, the perturbation scale remains fixed once hyperparameter values are chosen; on the other hand, the scale of the cost vector may increase or decrease throughout training. Consequently, the relative scale between perturbations and cost vectors can shift, leading to detrimental outcomes such as the collapse of learning into mere imitation or even complete degeneration.

For example, in the case of the DPO approach, the ML model may increase the norm of the cost vector at each epoch to gradually reduce the impact of perturbations on the estimated decision. This behavior is understandable, as it enables the model to stabilize around local minima. However, an excessive increase in cost vector norm prevents the model from identifying better cost vector mappings outside that local region. Therefore, it is necessary to regulate the scale of the cost vector to ensure optimal performance. We propose to address this issue by introducing a regularization of the cost vector estimate to control the stability of the optimization mapping $f$ in its neighborhood.

Consider a perturbed optimization mapping $\tilde{f} : \Theta \mapsto \mathcal{C}$ as defined in equation 9. Our proposed regularization is applied to the exact cost vector estimate $\boldsymbol{\theta}$ before cost perturbation. It is defined as a mapping $r : \Theta \to \Theta$ satisfying $f(r(\boldsymbol{\theta})) = f(\boldsymbol{\theta})$, so that the exact decision mapping remains unchanged. The role of $r$ is to rescale $\boldsymbol{\theta}$ to match the scale of the perturbation $\boldsymbol{\delta}$ introduced by the perturbed optimization mapping $\tilde{f}$, such that $\tilde{f}(r(\boldsymbol{\theta})) = f(r(\boldsymbol{\theta}) + \boldsymbol{\delta})$. The regularized decision-focused learning model is then described by:

$$\boldsymbol{\theta} = h_{\boldsymbol{v}}(\boldsymbol{x}), \quad \boldsymbol{y} = \tilde{f}\big(r(\boldsymbol{\theta})\big) \tag{20}$$

Managing solution stability to improve the reliability of the learning process while maintaining performances appears to be a significant challenge. According to property 2, it is necessary that the stability radius of the cost vector estimate be on the same scale as that of the perturbation to ensure this outcome. To the best of our knowledge, no further information is available regarding the exact value that the stability radius should assume to maximize the efficiency of the differentiation process. We therefore propose imposing bounds on the stability radius of all estimates in order to control its scale without assigning an exact value. According to property 3, the stability radius of a cost vector is directly proportional to its norm since it is a natural measure of its scale. Hence imposing bounds on the vector norm should constrain the stability radius. It is indeed the case for upper-bounds, as demonstrated by the following property:

**Property 5** *Given a non-constant optimization mapping $f : \Theta \to \mathcal{C}$, the stability radius of each cost vector in a bounded subset $\mathbb{T} \subset \Theta$ is uniformly upper-bounded.*

A proof of this property is provided in annex B. However, there does not exist a similar general property on lower bounds with regards to the relation between vector norm and stability radius, as shown by the counter-example provided in annex A. Instead and as illustrated in annex A, it appears that the subset of cost vectors with lower-bounded stability radius is dependent on the optimization problem structure. We restrict our work to general regularization approaches dedicated to the correct

management of solution stability in decision-focused learning, hence we focus in what follows on regularization to impose upper bounds on vectors norm and stability radius. Nonetheless, the development of regularization techniques to enforce lower bounds on the stability radius remains an open question for future research.

Our first approach consists in a $L^2$ normalization of the cost vector estimate, such that $r^n(\theta) = \frac{\theta}{\|\theta\|}$. Given the scale invariance of the optimization mapping, one has $f(r^n(\theta)) = f(\theta)$. We project the cost vector estimates unto the unit sphere, thereby upper-bounding the value of their stability radius. Note that in this first approach, for any cost vector $\theta \in \Theta$, the stability radius of $\alpha \cdot \theta$ is constant for all positive scalar $\alpha$. In other words, for a given direction component of a cost vector, the stability radius of its regularized counterpart is constant regardless of its scale. Hence, although this regularization does not assign a single value to the stability radius of all cost vectors, it does so for vectors sharing the same direction component. This partly contradicts our assumption that the stability radius should remain flexible in its precise value.

We propose to consider another regularization in which the cost vector norm is upper-bounded rather than having its value set to $1$, thereby bounding the stability radius for a given direction component rather than assigning it an arbitrary value. Our alternative approach consists in smoothly rescaling cost vector estimates so that their $L^2$ norm does not exceed a given positive value $\kappa$:

$$r^p(\boldsymbol{\theta}) = \frac{1}{1 + \kappa^{-1}\|\boldsymbol{\theta}\|} \cdot \boldsymbol{\theta} \tag{21}$$

Given the scale invariance of the optimization mapping, one has $f(r^p(\boldsymbol{\theta})) = f(\boldsymbol{\theta})$. This regularization corresponds to a projection into an $L^2$ ball of radius $\kappa$ ensuring the intended balance between rescaling and flexibility outlined above. It preserves the ordering of vector norms, leaving cost vectors with norms well below $\kappa$ essentially unchanged while down-scaling those with larger norms. The choice of value for parameter $\kappa$ balances flexibility and regularization: larger values yield looser bounds on both the norm and the stability radius.

# 6 NUMERICAL EXPERIMENTS

We run experiments to demonstrate the relevance of the proposed regularization approach. Note that it appears hardly possible to design an experimental setup in which training instability due to perturbations in DFL approaches could be reliably measured. Indeed, such instability is neither systematic nor easily identifiable. In other words, one cannot know a priori whether a DFL model will exhibit training instability, nor can one generate cases subject to such instability without introducing unrealistic or grotesque conditions.

We therefore propose a rigorous study based on an experimental setup using optimization problems and datasets drawn from the state-of-the-art review by Mandi et al. (2024). We run experiments that consists in three combinatorial optimization problems originating from other works in the DFL literature with publicly available datasets, each with three instances: the synthetic shortest path problem (SP), the set matching problem (SM), and the knapsack problem (K). We aim to evaluate the relevance of regularization for the smart predict-then-optimize loss, the implicit differentiation by perturbation and the differentiable perturbed optimizers. To this end, the performance of the SPO, DBB, and DPO approaches is evaluated in terms of regret expressed as a percentage, both with and without regularization. For each instance of each problem, we consider a learning model as defined by Mandi et al. (2024): it can be described by equation 2 such that $h_{\boldsymbol{v}}(\cdot)$ is a fully connected neural network. Hence the considered loss function for the DPO and DBB approach is the regret loss $L^r$ defined in equation 4. For context and comparison, we also report the results from predictions obtained with a standard mean-squared error (MSE) loss.

The experimental setup proposed by Mandi et al. (2024) is replicated in this study. For each experiment, ten trials are conducted during both validation and testing phases, with network weights initialized using distinct seeds ranging from 0 to 9. Performance is assessed based on the average regret over the test dataset across these ten training runs. The number of training epochs is fixed at 30, and a grid search is performed to identify the optimal hyper-parameter set for each DFL technique using a designated evaluation dataset. Details about the experimental setup, including the range of hyper-parameter values, can be found in annex C. The code is directly derived from the original work of Mandi et al. (2024) and is available for reproducibility purposes (Anonymous, 2025).

Table 1: Regret as percentage (%) for various problems and DFL techniques.

| | MSE | SPO | SPO-$r^n$ | SPO-$r^p$ | DPO | DPO-$r^n$ | DPO-$r^p$ | DBB | DBB-$r^n$ | DBB-$r^p$ |
|---|---|---|---|---|---|---|---|---|---|---|
| SP1 | 15.29 | 21.24 | **15.40** | **15.40** | 17.73 | **16.08** | 16.30 | 16.55 | *17.92* | **16.41** |
| SP2 | 10.12 | **10.07** | *10.12* | *10.40* | 11.83 | **10.55** | 11.18 | 11.92 | 11.32 | **10.83** |
| SP3 | 19.79 | 8.43 | **7.67** | 8.50 | 125.57 | **13.10** | 15.72 | 10.15 | **9.63** | *11.29* |
| SM1 | 92.09 | 91.72 | 90.38 | **90.09** | 92.08 | *92.27* | **92.03** | 91.53 | 91.16 | **90.62** |
| SM2 | 92.23 | 91.90 | **89.31** | 89.66 | 92.35 | *92.66* | **92.33** | 91.19 | 90.90 | **90.17** |
| SM3 | 91.87 | **88.02** | *88.79* | 88.14 | 92.35 | 92.33 | **92.05** | 90.19 | *91.13* | **88.97** |
| K1 | 14.06 | 11.92 | **11.85** | 11.89 | 14.30 | *16.24* | **14.16** | 10.88 | 10.85 | **10.80** |
| K2 | 7.66 | **5.44** | *5.84* | *5.50* | 6.06 | **5.53** | 5.66 | 5.12 | *5.28* | **5.12** |
| K3 | 4.01 | 2.46 | **2.43** | **2.43** | 3.40 | *3.8* | **3.40** | 2.34 | *2.57* | **2.34** |

Table 1 presents the performance, in terms of regret, of each DFL technique under each regularization method and for each problem, using their respective best hyper-parameter configurations. The best results for each DFL technique and optimization problem are shown in **bold**, while regularized DFL techniques performing worse than their unregularized counterparts are shown in *italics*.

The regularizations $r^n$ and $r^p$ outperform the standard DFL techniques in 15 and 22 out of 27 cases, respectively. In scenarios where both regularizations improve performance over the standard DFL technique, neither shows a significant advantage over the other. There exist a few cases where regularization does not outperform the standard DFL techniques. For regularization $r^n$ especially, it could be explained by the lack of flexibility in the possible values taken by the stability radius. Moreover for both regularizations, the learning process may sometimes benefit from converging to local minima with large-norm vectors. For the SPO approach especially, this aligns with imitation-based learning occasionally outperforming experience-based learning. Even so, projection-based regularization $r^p$ appears more reliable than normalization-based regularization $r^n$, as it consistently outperforms all considered DFL techniques. We therefore conclude that introducing a regularization strategy that balances rescaling and flexibility is essential for consistently improving perturbation-based DFL techniques.

Remember that we do not know a priori whether some of the considered problems and datasets are subject to instability during DFL training. In cases where instability does not occur, regularization provides little benefit, which explains the marginal improvements observed for certain problems and DFL techniques. Nonetheless, for problems where instability in DFL training is prevalent, regularization clearly enhances performance.

# 7 CONCLUSION

In decision-focused learning, a key challenge arises from the solution stability of discrete optimization mappings when subjected to additive perturbations. Using theoretical insights from stability analysis in combinatorial optimization, we formally characterized the response of these mappings to such perturbations by study of their stability radius. Our analysis revealed that perturbation-based DFL methods can become inefficient when the estimated cost vector and the perturbations are not properly scaled relative to one another. To address this, we proposed regularizing the cost vector estimates during training. By striking a balance between rescaling and flexibility, our designed regularization approach consistently improved all studied DFL techniques across a large variety of optimization problems and datasets.

Our work paves the way for multiple directions of future research. Our study on solution stability was limited to additive perturbations, as they are the most prevalent in DFL methods. However, alternative forms of perturbation may influence the differentiation process in other ways. For instance, multiplicative perturbation for DFL techniques (Dalle et al., 2022) might be promising in addressing issues related to solution stability, since perturbations in that case are at scale with cost vector estimates. Our study focused on regularization methods that set an upper bound on the stability radius, while outlining the challenges of doing the same for lower bounds. Exploring how the problem's structure influences solution stability could help address these difficulties.

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

## A  EXAMPLE TO ILLUSTRATE SOLUTION STABILITY

Consider the following problem in two dimensions, with linear costs parameterized by a vector $\boldsymbol{\theta} \in \mathbb{R}^2$, and denoted $P(\boldsymbol{\theta})$:

$$P(\boldsymbol{\theta}) : \min \ \boldsymbol{\theta}_1 \cdot \boldsymbol{x}_1 + \boldsymbol{\theta}_2 \cdot \boldsymbol{x}_2 \tag{22}$$

$$\boldsymbol{x}_1 + \boldsymbol{x}_2 = 1 \tag{23}$$

$$0 \leq \boldsymbol{x}_i \leq 1, \qquad\qquad \forall i \in \{1, 2\} \tag{24}$$

In this simple problem, the optimal solution depends entirely on the order of costs. Indeed, the optimal solution is $\boldsymbol{x} = (1, 0)$ (resp. $\boldsymbol{x} = (0, 1)$) for all cost vectors such that $\boldsymbol{\theta}_1 < \boldsymbol{\theta}_2$ (resp. $\boldsymbol{\theta}_1 > \boldsymbol{\theta}_2$). Otherwise for all cost vectors such that $\boldsymbol{\theta}_1 = \boldsymbol{\theta}_2$, any solution $\boldsymbol{x} \in \{(a, 1 - a); \ a \in [0, 1]\}$ is optimal.

Consider the set valued optimization mapping $F(\cdot)$ associated to problem $P$. The set $F(\boldsymbol{\theta})$ is a singleton $\{(1, 0)\}$ (resp. $\{(0, 1)\}$) for all cost vectors such that $\boldsymbol{\theta}_1 < \boldsymbol{\theta}_2$ (resp. $\boldsymbol{\theta}_1 > \boldsymbol{\theta}_2$), and is equal to $\{(a, 1 - a); \ a \in [0, 1]\}$ for all cost vectors such that $\boldsymbol{\theta}_1 = \boldsymbol{\theta}_2$. Hence $F(\boldsymbol{\theta})$ is a singleton for all cost vectors $\theta \in \mathbb{R}^2$ except for those with a direction component along vector $(1, 1)$. We conclude that $F(\cdot)$ is a singleton almost everywhere.

Consider now the optimization mapping $f : \mathbb{R}^2 \mapsto [0, 1]^2$ associated to problem $P$, such that $f(\boldsymbol{\theta})$ is an optimal solution to problem $P(\boldsymbol{\theta})$ for a cost vector $\boldsymbol{\theta} \in \mathbb{R}^2$. One can observe the invariance property for this particular example. For all cost vectors $\boldsymbol{\theta} \in \mathbb{R}^2$ and positive scalars $\alpha > 0$, one has $f(\alpha \cdot \boldsymbol{\theta}) = f(\boldsymbol{\theta})$ since the order of costs remains the same for vectors $\boldsymbol{\theta}$ and $\alpha \cdot \boldsymbol{\theta}$.

We now determine an explicit formulation to compute the stability radius of any cost vector for that particular problem. To that end, consider a cost vector $\boldsymbol{\theta} \in \mathbb{R}^2$ such that $\boldsymbol{\theta}_1 > \boldsymbol{\theta}_2$ and a perturbed cost vector $\boldsymbol{\theta} + \boldsymbol{\delta} \in \mathbb{R}^2$. The order of costs is identical for these two vectors if the $L^2$ norm of $\boldsymbol{\delta}$ is lower than $\boldsymbol{\theta}_1 - \boldsymbol{\theta}_2$, such that the first element of $\boldsymbol{\theta} + \boldsymbol{\delta}$ cannot get smaller than the second element of that perturbed cost vector. Likewise, suppose that $\boldsymbol{\delta} = (0, \boldsymbol{\theta}_1 - \boldsymbol{\theta}_2 + \epsilon)$ for a given positive scalar $\epsilon > 0$. This vector is of $L^2$ norm $\boldsymbol{\theta}_1 - \boldsymbol{\theta}_2 + \epsilon$, and one has $\boldsymbol{\theta} + \boldsymbol{\delta} = (\boldsymbol{\theta}_1, \boldsymbol{\theta}_1 + \epsilon)$, hence the orders of cost of $\boldsymbol{\theta}$ and $\boldsymbol{\theta} + \boldsymbol{\delta}$ are different. We deduce that $\boldsymbol{\theta}$ and $\boldsymbol{\theta} + \boldsymbol{\delta}$ are of different orders of costs if and only if $\boldsymbol{\delta}$ is of greater norm than $\boldsymbol{\theta}_1 - \boldsymbol{\theta}_2$.

A symmetrical result can be found for a cost vector $\boldsymbol{\theta} \in \mathbb{R}^2$ such that $\boldsymbol{\theta}_1 < \boldsymbol{\theta}_2$, such that $\boldsymbol{\theta}$ and $\boldsymbol{\theta} + \boldsymbol{\delta}$ are of different orders of costs if and only if $\boldsymbol{\delta}$ is of greater norm than $\boldsymbol{\theta}_2 - \boldsymbol{\theta}_1$. Since the order of costs for any vector $\boldsymbol{\theta}$ determines its optimal solution $f(\boldsymbol{\theta})$ for problem $P$, we conclude that the stability radius of optimization mapping $f(\cdot)$ at $\boldsymbol{\theta}$ is equal to $|\boldsymbol{\theta}_1 - \boldsymbol{\theta}_2|$.

Consider now the sequence of cost vectors $\left((t + \frac{1}{n}, t)\right)_{n>1}$ for a positive scalar $t > 0$. All cost vectors in that sequence have an $L^2$ norm greater than $t$. The associated stability radius sequence is $(\frac{1}{n})_{n>1}$, hence it converges to zero, independently of the scalar $t$. It follows in this particular case that, no matter the scale, there exist cost vectors in $\mathbb{R}^2$ with a stability radius of arbitrarily low value. This serves as a counter-example to the hypothesis, formulated in section 5, that a positive lower bound on the norm of cost vectors implies a positive lower bound on the associated stability radius. In other words, even if the norm of the cost vectors is bounded below, it does not guarantee the existence of a positive lower bound for the stability radius.

In this particular case, the subset of cost vectors in $\mathbb{R}^2$ with a lower bound $\underline{\kappa}$ on their stability radius is defined by $\{\boldsymbol{\theta} \in \mathbb{R}^2; |\boldsymbol{\theta}_1 - \boldsymbol{\theta}_2| \geq \underline{\kappa}\}$. We conclude that a subset of cost vectors with a lower bound on their stability radius can be identified by analysis of the optimization problem structure.

## B  PROOF OF PROPERTY 5

We provide a proof for property 5.

First, we denote by $\mathcal{B}(\boldsymbol{\theta}, \rho)$ the $L^2$ ball of radius $\rho$ in $\mathbb{R}^n$ centered on $\boldsymbol{\theta}$. We consider a non-constant optimization mapping $f : \Theta \mapsto \mathcal{C}$. Let $\mathbb{T} \subset \Theta$ be a bounded set, with $\rho$ an upper-bound on the norm of all cost vectors in that set. If there does not exist an upper-bound on the stability radius of each cost vector in that set, then there exists a cost vector $\boldsymbol{\theta}^0 \in \mathbb{T}$ such that its stability radius is greater than $2\rho$. Hence for all perturbation $\boldsymbol{\delta} \in \Theta$ with a norm lower than $2\rho$, one has $f(\boldsymbol{\theta}^0 + \boldsymbol{\delta}) = f(\boldsymbol{\theta}^0)$.

Note that the set $\mathcal{B}(\mathbf{0}, \rho)$ is a subset of $\mathcal{B}(\boldsymbol{\theta}^0, 2\rho)$ since $\boldsymbol{\theta}^0 \in \mathcal{B}(\mathbf{0}, \rho)$. We therefore conclude that one has $f(\boldsymbol{\theta}) = f(\boldsymbol{\theta}^0)$, for all $\boldsymbol{\theta} \in \Theta \cap \mathcal{B}(\mathbf{0}, \rho)$. Hence according to property 3, the optimization mapping $f$ is constant over the set $\Theta$. This is absurd since the optimization mapping $f$ is supposed to be non-constant, hence we conclude that the stability radius is uniformly upper-bounded over the bounded set $\mathbb{T} \subset \Theta$.

## C  DETAILED EXPERIMENTATION SETUP

We propose in table 2 the value range of each parameter involved in one of the studied learning approaches presented in section 6.

| Hyper-parameter | DFL technique | Range |
|:---:|:---:|:---:|
| learning rate | All | $\{10^{-4}, 10^{-3}, 10^{-2}, 10^{-1}, 1\}$ |
| $\kappa$ | All | $\{1, 10^2, 10^4\}$ |
| $\alpha$ | SPO | $\{0.5, 2, 10\}$ |
| $\beta$ | DBB | $\{0.1, 1, 10\}$ |
| $\epsilon$ | DPO | $\{0.1, 0.5, 1\}$ |

Table 2: Value table for the hyper-parameter tuning of all learning approaches

All parameters range have been selected according to pre-tests leading to the dismissal of exceedingly large or small values, based on numerical experiments proposed by Mandi et al. that designed this benchmark. Regarding the range of values for the parameter $\kappa$ in regularization $r^p$, they act as a measure of the intensity of the regularization, with each value corresponding respectively to a strict, moderate or low regularization. The number of possible values for each parameter range is assumed to be fair given the extensive computation time required to train each decision-focused learning model.

All experiments and results detailed in section 6 can be reproduced with the code provided in the indicated Git repository (Anonymous, 2025).

## D  LLM DECLARATION OF USE

A Large Language Model (GPT-5) was used to polish writing and identify relevant research works in perturbation analysis literature.

