# OpenReview forum: "Managing Solution Stability in Decision-Focused Learning with Cost Regularization"
_ICLR.cc/2026/Conference — Submitted to ICLR 2026_

### Official Review · Reviewer_xxyc · 2025-10-26

**Soundness:** 3
**Presentation:** 3
**Contribution:** 2
**Rating:** 2
**Confidence:** 4

**Summary:**

This paper analyzes gradient estimators for decision-focused learning/differentiation through discrete optimization and provides theoretical arguments that the "scale" of extra ingredients in these estimators (e.g., random perturbations) should be set appropriately relative to the scale of coefficients in the objective function.

**Strengths:**

Stability is a real issue in any attempt to include differentiable optimization in machine learning pipelines. Existing pipelines are prone to uneven performance depending on a number of hyperparameter choices. It makes sense that normalizing the scale of the coefficients in the optimization problem could be useful to help deal with some of these issues, or make setting other hyperparameters easier.

**Weaknesses:**

The bulk of the paper is spent on theoretical analyses that do not add a great deal of information compared to what is already known. It is not surprising that when the hyperparameters of existing methods are set on an inappropriate scale, their gradient estimators would no longer be tracking a useful quantity. The question is more (a) how to remedy such issues, and (b) if doing so resolves some of the overall stability issues in training with optimization in the loop. The paper makes a couple of suggestions about (a) , but it's not clear if this is addressing a large issue in the overall pipeline, or whether training instability is largely a result of other characteristics besides setting the right scale for these hyperparameters. The empirical results don't show a very consistent or large improvement from any particular strategy compared to unmodified existing approaches. Even when there is an improvement, it is often very slight (not really practically significant, and unclear whether statistically significant).

**Questions:**

NA

---

> ### Author Response · Authors · 2025-11-14
> **Answers to Reviewer xxyc**
>
> We thank the reviewer for the time and effort invested in the evaluation of our work. We propose to answer all questions and issues pointed out in their report.
>
> **Q: The bulk of the paper is spent on theoretical analyses that do not add a great deal of information compared to what is already known.**
>
> A:  We agree with the reviewer that some observations may have already been made in the DFL community regarding the scale of perturbation compared to that of the cost vector estimate in DFL models. However, it has not yet been formalized until now to the best of our knowledge. Our first contribution to the theoretical analysis in this article is the formalization of a principled connection between perturbation analysis, solution stability, and the proper handling of perturbations in decision-focused learning. In turn, this new formalism enables the design of appropriate regularization strategies, supported by rigorous proofs of their relevance, as exemplified in property 5. More generally, this theoretical analysis provides essential insights into more effective perturbation management and lays the groundwork for future research directions.
>
> The detailed examination of each perturbation-based DFL technique reveals the direct impact of mismanaging perturbations on the learning process. In particular, it clarifies why this issue may have remained overlooked in cases where learning collapses into imitation, and why results often appear lackluster when the learning process degenerates entirely. Hence this contribution is valuable from a practical standpoint as it explains unexpected behaviors observed within DFL pipeline in training phase. Moreover it motivates the need for regularization strategies since it is revealed, as discussed below, that learning may degenerate if cost vectors scale are left unchecked.

---

> ### Author Response · Authors · 2025-11-14
> **Second part of answers to Reviewer xxyc**
>
> **Q: It is not surprising that when the hyperparameters of existing methods are set on an inappropriate scale, their gradient estimators would no longer be tracking a useful quantity. The question is more (a) how to remedy such issues, and (b) if doing so resolves some of the overall stability issues in training with optimization in the loop. The paper makes a couple of suggestions about (a) , but it's not clear if this is addressing a large issue in the overall pipeline, or whether training instability is largely a result of other characteristics besides setting the right scale for these hyperparameters.**
>
> A: **The comments suggest there may be some misunderstanding regarding the purpose of our regularization approach, which is a central contribution of the paper. We would like to clarify that point, both in this comment and in the article.**
>
> We agree with the reviewer that setting hyperaparameters scale is not sufficient to address training instability. While setting hyperparameters on an appropriate scale is necessary to ensure that gradient estimates remain meaningful (as noted in lines 361–363), this alone does not prevent the adverse effects of perturbation mismanagement during learning, as later argued in the same paragraph. **This is the exact reason why we introduce regularization strategies: addressing a large issue in the overall pipeline throughout learning which is a result of other characteristics besides setting the right scale for these hyperparameters .**
>
> Indeed, hyperparameters may be set to values such that solution stability may initially be under control in the training process. However, the weights of the ML model and its outputs evolve over the learning process, potentially altering their scale. On one hand, the perturbation scale remains fixed once hyperparameter values are chosen; on the other hand, the scale of the cost vector may increase or decrease throughout training. Consequently, the relative scale between perturbations and cost vectors can shift, leading to detrimental outcomes such as the collapse of learning into mere imitation or even complete degeneration.
>
> For example in the case of the DPO approach and no matter the intensity of perturbation, the ML model may increase the norm of the cost vector at each epoch to gradually reduce the impact of the perturbation on the estimated decision. This behavior is understandable, as it enables the model to stabilize around local minima. However, an excessive increase in cost vector norm prevents the model from identifying better cost vector mappings outside that local region. Therefore, no matter the hyperparameters, it is necessary to regulate the scale of the cost vector to ensure optimal performance.
>
> We therefore advocate regularizing the cost vectors so that they consistently remain at scale with the perturbations, regardless of the values taken by the model's weights and outputs throughout the entire learning process. More details are now provided in the beginning of section 5 to better highlight the purpose of our regularization strategy.
>
> Finally, we contend that the proposed regularization approaches are more than mere suggestions. Their relevance is firmly grounded in the theoretical analysis of solution stability presented in section 3.2 and further supported by property 5, which derives directly from this analysis. Without the theoretical analysis, there would have been no ground for such approaches. In turn, the proposed regularization strategies directly address this issue with broad applicability, irrespective of the optimization mapping, thanks to the theoretical result established in property 5.
>
> **Q: It makes sense that normalizing the scale of the coefficients in the optimization problem could be useful to help deal with some of these issues, or make setting other hyperparameters easier.**
>
> A: A benefit of establishing a connection between solution stability and decision-focused learning lies in the discussion in section 5 regarding what kind of normalization would be adequate in a DFL pipeline. First, we actually prove that normalization could be beneficial in managing the stability radius. Then, we highlight that a straightforward normalization may be detrimental, hence we introduce and alternative projection-based approach. Those theoretical and practical results are more precise and grounded than a merely intuitive idea of what could help deal with some of these issues, and paves the way for future research work on the topic.

---

> ### Author Response · Authors · 2025-11-14
> **Third part of answers to Reviewer xxyc**
>
> **Q:  The empirical results don't show a very consistent or large improvement from any particular strategy compared to unmodified existing approaches. Even when there is an improvement, it is often very slight (not really practically significant, and unclear whether statistically significant**
>
> A: We agree that the empirical results do not show a dramatic improvement for all cases. That is to be expected given the experimental setup we decided to consider with appropriate rigorousness.
>
> We point out that it appears hardly possible to design an experimental setup in which training instability due to perturbations in DFL approaches could be reliably measured. Indeed, such instability is neither systematic nor easily identifiable. In other words, one cannot know a priori whether a DFL model will exhibit training instability, nor can one generate cases subject to such instability without introducing unrealistic or grotesque conditions.
>
> We therefore propose a rigorous study based on an honest experimental setup using optimization problems and datasets drawn from the state-of-the-art review by Mandi et al. (2024). We do not know a priori whether some of these problems and datasets are subject to instability during DFL training. In cases where instability does not occur, regularization provides little benefit, which explains the marginal improvements observed for certain problems and DFL techniques. Nonetheless, for problems where instability in DFL training is prevalent, regularization clearly enhances performance. The practical significance of our contribution lies in the proper management of instability in cases where it severely hinders the DFL learning process.

---

> ### Author Response · Authors · 2025-11-25
> **Official Comment by Authors**
>
> We sincerely appreciate the reviewers' valuable feedback. We have carefully addressed the concerns raised and hope that our responses have provided sufficient clarification. If you have any questions that require further discussion, we are happy to discuss.
>
> Could you please let us know whether our rebuttal resolves your concerns? We look forward to your further comments and feedback.

---

### Official Review · Reviewer_5AaY · 2025-10-31

**Soundness:** 1
**Presentation:** 3
**Contribution:** 2
**Rating:** 2
**Confidence:** 4

**Summary:**

This paper studies the impact of solution stability on the convergence of the DFL training process. The authors consider a classical DFL setting, which involves a ML estimator used to output the cost function coefficients for an optimization problem with a fixed feasible space.

They focus on methods that work by perturbing the estimated parameters and the impact of such perturbation on the decision problem solution. Scale has a key role here: too small perturbations do not cause the solution to change and hence provide no useful information for gradient descent; too large perturbations lead to radical changes of the optimal solution and risk causing instability.
The paper also explains how a poorly chosen scale can make a number of "experience based" DFL method to become equivalent to (usually less performant) "imitation based" methods.

The authors then propose to address the issue by applying a mapping to the predicted parameters (perturbed on unaltered) that normalizes the scale of the vector. In a compact empirical evaluation, the method proves capable of improving the performance of some DFL techniques in some cases

**Strengths:**

I think this work makes a very good case for how controlling the scale of perturbations (or of sampling processes) can dramatically affect the behavior of DFL training methods that makes use of such idea (which are at this point many and among the best performers).

The discussion on how different classes of method become either ineffective, or collapse to solution imitation, is well done and convincing, even if somewhat informal.

I also believe that the proposed normalization technique can be useful in the considered setting, by far the most common in the DFL literature and involving decision problems with linear costs.

**Weaknesses:**

The key issue I see in this work is that the proposed approach does not appear to address the analyzed problem. Based on the formulation from eq. (19), the normalization mapping is applied to the parameter vector just before it is fed to the optimization process (the f mapping). In a perturbation based approach, this means that normalization would be applied to the perturbed parameters, after the scale mismatch as already done all the damage extensively documented in the first half of the paper.

The proposed normalization seems instead well suited to address a second, more widely acknowledged, issue in the considered setting, i.e. the fact that the optimization mapping is scale-invariant (as stated by the authors in Property 3).

Overall, the analysis the take most of the paper is devoted to one problem, but the proposed mitigation and the empirical evaluations are about another, different, problem.

As a secondary, but still significant, weakness, the reported improvements are not particularly large in all but one or two cases.

The remaining issues I could find are minor. Here are some of them:

* The lack of differentiability is not the real issue in the considered setting (the solution function is differentiable almost everywhere). The true problem is that the gradient is 0, and therefore non-informative from an optimization standpoint, whenever it is defined. There's confusion on this point in several places in the paper (but this is easy to fix)
* The rationale for the g mapping is extremely unclear, especially since it appears to be virtually ignored in the entire paper, including the empirical evaluation (where it is explicitly an identity function)
* The g mapping also appears to have discrete output, meaning it cannot be differentiable as stated at line 115
* The first proposed normalization maps the parameter vectors onto the surface of the unit sphere, rather than into the unit sphere (this is a good thing, actually)
* Several of the considerations at lines 169-175 appear to be true by construction, and hence do not see to add much to the discussion

**Questions:**

* Is the normalization mapping actually applied to the cost vector just before the optimization mapping?
* Why is the g mapping included in the formulation?

---

> ### Author Response · Authors · 2025-11-14
> **Answers to Reviewer 5AaY**
>
> We thank the reviewer for the time and effort invested in the evaluation of our work. We propose to answer all questions and issues pointed out in their report.
>
> **Q: Overall, the analysis the take most of the paper is devoted to one problem, but the proposed mitigation and the empirical evaluations are about another, different, problem. Is the normalization mapping actually applied to the cost vector just before the optimization mapping?**
>
> A: **The comments suggest there may be some misunderstanding regarding the purpose of our regularization approach, which is a central contribution of the paper. We would like to clarify that point, both in this comment and in the article.**
>
> We strongly insist on the fact that regularization  is not applied to the cost vector just before the optimization mapping. As the reviewer points out, that would be hardly relevant. Instead, we regularize the exact cost vector estimate **before** perturbation is applied, so that both cost vector and perturbation are on the same scale. We apologize for the confusion.
>
> The purpose of the introduced regularization is to address the one problem presented in the theoretical part of the article, that is the difference in scale between the cost vector estimate and the perturbation. Hence we propose to regularize the exact cost vector estimate before perturbation is applied. That way, we ensure that the cost vector estimate is at scale with the perturbations.
>
> Indeed, hyperparameters may be set to values such that solution stability may initially be under control in the training process. However, the weights of the ML model and its outputs evolve over the learning process, potentially altering their scale. On one hand, the perturbation scale remains fixed once hyperparameter values are chosen; on the other hand, the scale of the cost vector may increase or decrease throughout training. Consequently, the relative scale between perturbations and cost vectors can shift, leading to detrimental outcomes such as the collapse of learning into mere imitation or even complete degeneration.
>
> For example in the case of the DPO approach and no matter the intensity of perturbation, the ML model may increase the norm of the cost vector at each epoch to gradually reduce the impact of the perturbation on the estimated decision. This behavior is understandable, as it enables the model to stabilize around local minima. However, an excessive increase in cost vector norm prevents the model from identifying better cost vector mappings outside that local region. Therefore, no matter the hyperparameters, it is necessary to regulate the scale of the cost vector to ensure optimal performance.
>
> Our proposed regularization is applied to the exact cost vector estimate $\theta$ before cost perturbation. Consider a perturbed optimization mapping $\tilde{f}: \Theta \mapsto \mathcal{C}$. Then regularization takes the form of a mapping $r:\Theta \mapsto \Theta$,that does not impact the exact decision mapping, i.e. such that $f(r(\theta))=f(\theta)$, but that rescales vector $\theta$ so that it matches the scale of perturbations applied by the perturbed optimization mapping $\tilde{f}$. The regularized decision-focused learning model is then described by:
> \begin{equation}
> \theta = h_{v}(x), \quad y=\tilde{f}\big(r(\theta)\big), \quad z= g_{u}(y)
> \end{equation}
>
> Details are provided in the beginning of section 5 to better introduce our regularization strategy.
>
> **Q: As a secondary, but still significant, weakness, the reported improvements are not particularly large in all but one or two cases.**
>
> A: We agree that the empirical results do not show a dramatic improvement for all cases. That is to be expected given the experimental setup we decided to consider with appropriate rigorousness.
>
> We point out that it appears hardly possible to design an experimental setup in which training instability due to perturbations in DFL approaches could be reliably measured. Indeed, such instability is neither systematic nor easily identifiable. In other words, one cannot know a priori whether a DFL model will exhibit training instability, nor can one generate cases subject to such instability without introducing unrealistic or grotesque conditions.
>
> We therefore propose a rigorous study based on an honest experimental setup using optimization problems and datasets drawn from the state-of-the-art review by Mandi et al.~(2024). We do not know a priori whether some of these problems and datasets are subject to instability during DFL training. In cases where instability does not occur, regularization provides little benefit, which explains the marginal improvements observed for certain problems and DFL techniques. Nonetheless, for problems where instability in DFL training is prevalent, regularization clearly enhances performance.
>
> The practical significance of our contribution lies in the proper management of instability in cases where it severely hinders the DFL learning process.

---

> > ### Comment · Reviewer_5AaY · 2025-11-19
> > **Thanks for the clarification**
> >
> > As the title says: applying the normalization _before_ the perturbation makes perfect sense. This resolve my main concern and I will update my score accordingly.
> >
> > I should however point out that the notation in eq (19), even in the current version of the paper does not match what you are doing. I suggest that you change the equation so that it explicitly shows the perturbation term: that will make it explicit that the mapping is applied only to the predictions.
> >
> > Concerning the improvements, I understand the difficulty, though the limited improvement remains a point of weakness.
> >
> > If you follow up to this line of work, it might be worth thinking of a controlled experiment: this could be done by either crafting a toy problem with controllable scale issues, or by modifying an existing problem. Though obviously less realistic, that would allow you to study in-depth the impact of scale mismatch.

---

> ### Author Response · Authors · 2025-11-14
> **Second part of answers to Reviewer 5AaY**
>
> **Q: The lack of differentiability is not the real issue in the considered setting (the solution function is differentiable almost everywhere). The true problem is that the gradient is 0, and therefore non-informative from an optimization standpoint, whenever it is defined. There's confusion on this point in several places in the paper (but this is easy to fix)**
>
> A: Indeed, we apologize for the confusion regarding the differentiability of the optimization mapping $f$. In most cases, $f$ is piecewise constant, which implies that its gradient is zero almost everywhere. Since this gradient has limited relevance in an optimization context, DFL techniques are employed to identify meaningful descent directions for minimizing the loss function. The article has been revised to better reflect this particular characteristic.
>
> **Q: Several of the considerations at lines 169-175 appear to be true by construction, and hence do not see to add much to the discussion**
>
> A: Although this point does not significantly advance the flow of our discussion, it is necessary to ensure the rigor of our claim. We agree that it is most often admitted that the set-valued mapping $F$ is a singleton almost everywhere. However in our case, we draw upon specific findings from a body of literature outside machine learning—namely, perturbation analysis—and therefore particular care is taken in adapting these results to our context.
>
> **Q: The rationale for the g mapping is extremely unclear, especially since it appears to be virtually ignored in the entire paper, including the empirical evaluation (where it is explicitly an identity function). Why is the g mapping included in the formulation?**
>
> A: Indeed, the role of mapping $g$ deserves a more explicit introduction. The optimization mapping $ f : \Theta \mapsto \mathcal{C}$ can be incorporated at any layer of our DFL pipeline, rather than being restricted to the final stage. Mapping $g$ captures the possibility of integrating an additional ML component downstream of the decision mapping within the overall model. Most DFL approaches aim to predict the cost objective associated with the optimization mapping, as in our numerical experiments. In such cases,  $g$ typically reduces to the identity function. However, there are applications where this is not true, as illustrated in the work of (Rolinek et al., 2020).
>
> Although our examples and numerical experiments focus on this specific loss function and model structure, our contributions could generalize to all studied perturbation-based DFL techniques within the broader framework. We propose to better introduce mapping $g$ in the article to avoid confusion.

---

> > ### Comment · Reviewer_5AaY · 2025-11-19
> > **I get the point of the g mapping, but I still advise to keep it out**
> >
> > After your response, I now understand the point of the g mapping. I totally agree that the DFL "machinery" could be used as a building block for a more complex pipeline.
> >
> > However, scientifically speaking, it is bad practice to introduce something in the main formalization and then provide no evidence on how it can be used, what are its theoretical implications, and how it would affect the results.
> >
> > I would advise to remove the mapping from the notation in the main text, and keep just a mention that a second component could be integrated downstream. You could then formalize this point (using the current notation) and add an extensive discussion in the appendix. In this way, you could still get your point across, but without making a claim that is not backed by evidence.

---

> > > ### Author Response · Authors · 2025-11-20
> > > **Manuscript revision according to reviewer comments**
> > >
> > > We acknowledge the reviewer's concern that asserting our approach applies to more general pipelines may be overstated. Accordingly, we have revised the manuscript to remove the consideration of the mapping $g$, thereby focusing exclusively on the specific training pipeline associated with the regret loss.

---

> ### Author Response · Authors · 2025-11-20
> **Manuscript revision according to reviewer comments**
>
> We thank the reviewer for their valuable comment that help improve the paper.
>
> We propose the following revision regarding equation (19). We now define the perturbed decision mapping: perturbed decision mapping $\tilde{f}: \Theta \mapsto \mathcal{C}$ introduces an additive perturbation to the input cost vector before evaluation of the corresponding optimization mapping. In other word for a cost vector $\theta \in \Theta$ and a cost perturbation $\delta \in \Theta$, the corresponding perturbed decision is:
> $$
>     \tilde{f}(\theta) = f(\theta + \delta)
> $$
> Our regularization is then defined as a mapping $r: \Theta \to \Theta$ satisfying $f(r(\boldsymbol{\theta})) = f(\boldsymbol{\theta})$, so that the exact decision mapping remains unchanged. The role of $r$ is to rescale $\boldsymbol{\theta}$ to match the scale of the perturbation $\boldsymbol{\delta}$ introduced by the perturbed optimization mapping $\tilde{f}$, such that $\tilde{f}(r(\boldsymbol{\theta})) = f(r(\boldsymbol{\theta}) + \boldsymbol{\delta})$.
>
> Regarding the numerical experiments, it could indeed be valuable to evaluate our approach in a controlled environment, for instance through a toy problem that reproduces the negative impact of mismanaging solution stability during training. However, we consider that setting up such an experiment would require significant time; therefore, we do not propose to extend our numerical experiments in this direction during the rebuttal phase. We will however consider this possibility in future work.

---

### Official Review · Reviewer_8ggm · 2025-11-02

**Soundness:** 3
**Presentation:** 3
**Contribution:** 2
**Rating:** 6
**Confidence:** 3

**Summary:**

The paper focuses on decision-focused learning (DFL) and predicting coefficients in the objective function. The authors introduce the notion of solution stability and use this perspective to investigate different types of DFL techniques. They show that, in all examined perturbation-based DFL methods, solution stability around cost estimates is hard to predict, leading to a loss of learning signal or a shift from experience-based learning to imitation. To address this issue, they propose a regularization of the estimated cost vectors.

**Strengths:**

1. The writing is generally clear.
2. The viewpoint of interpreting different decision-focused learning methods through the concept of solution stability is novel and interesting.

**Weaknesses:**

1. The introduction and explanation of the four properties in Section 3 could be clearer; adding examples may aid understanding.
2. There are some typos—for example, inconsistent capitalization of the initial letter in “property.”

**Questions:**

1. DFL techniques are generally expected to outperform prediction-focused learning. However, according to Table 1, their performance is worse than that of the MSE training method, especially in the SP1 case. Could you explain this phenomenon?
2. Recently, there have been DFL studies that predict coefficients in the constraints. Can the notion of solution stability be used to analyze these works as well?

---

> ### Author Response · Authors · 2025-11-14
> **Answers to Reviewer 8ggm**
>
> We thank the reviewer for the time and effort invested in the evaluation of our work. We propose to answer all questions and issues pointed out in their report.
>
> **Q: The introduction and explanation of the four properties in Section 3 could be clearer; adding examples may aid understanding. There are some typos—for example, inconsistent capitalization of the initial letter in “property.”**
>
> A: The example provided in Annex A is now leveraged to illustrate properties 2 to 4. Typos are corrected, thank you for pointing them out.
>
> **Q: DFL techniques are generally expected to outperform prediction-focused learning. However, according to Table 1, their performance is worse than that of the MSE training method, especially in the SP1 case. Could you explain this phenomenon?**
>
> A: Indeed, in certain situations, standard prediction models can outperform decision-focused learning pipelines. This phenomenon is not unique to our case; Our experiments are identical to those in the state-of-the-art review by Mandi et al. (2024), in which similar patterns have been observed. While this topic merits deeper discussion, our position is that perturbation-based DFL techniques inherently involve a trade-off: they aim to improve decision quality at the expense of relying on approximate methods. When the drawbacks of this approximation outweigh its benefits, the resulting DFL model may underperform compared to more conventional approaches. This can be observed for example when the downstream optimization problem is robust to prediction errors (e.g., small changes in predictions don’t significantly affect the optimal decision), and the benefit of DFL is limited.
>
> **Q: Recently, there have been DFL studies that predict coefficients in the constraints. Can the notion of solution stability be used to analyze these works as well?**
>
> Indeed, the notion of solution stability fully applies to perturbations in constraint coefficients. In fact, Property 1, originating from the work of Bonnans and Shapiro (2000) and related to the upper semi-continuous property of the set-valued mapping, applies to additive perturbations in both the objective and the constraints. Likewise, the concepts of stability region and radius introduced by Sostskov (1992) also apply to perturbations in the objective as well as in the constraints. Hence, all results up to Property~3 can be extended to additive perturbations in constraints.
>
> However, the invariance described in Property 3 only holds for a parameterized objective. If all constraint coefficients were multiplied by a positive scalar, the resulting optimal decision could differ from that of the original optimization problem. Since all subsequent results in the remainder of the article rely on this property, it cannot be extended to additive perturbations in constraints.

---

> ### Author Response · Authors · 2025-11-25
> **Official Comment by Authors**
>
> We sincerely appreciate the reviewers' valuable feedback. We have carefully addressed the concerns raised and hope that our responses have provided sufficient clarification. If you have any remaining questions that require further discussion, we are happy to discuss.
>
> Could you please let us know whether our rebuttal resolves your concerns? We look forward to your further comments and feedback.

---

### Official Review · Reviewer_WL6A · 2025-11-10

**Soundness:** 2
**Presentation:** 2
**Contribution:** 2
**Rating:** 4
**Confidence:** 2

**Summary:**

This paper analyzes why perturbation-based Decision-Focused Learning (DFL) methods often become unstable or ineffective when training models to predict cost coefficients for MILPs.
The key insight is that the effectiveness of perturbation-based gradient signals depends on the relative scale between the learned cost vector and the perturbations.
If this scale is poorly matched, gradients either vanish or push the model toward pure imitation, resulting in performance collapse.

To address this, the paper introduces a cost regularization strategy that controls the stability radius of the optimization mapping, thereby improving the quality of the gradient signal. Two forms of regularization are studied:
- L2-normalization of cost vectors (rn)
- Projection into a bounded L2-ball with radius κ (rp)

Experiments show that regularizing cost vectors improves training stability and decision performance across multiple discrete optimization benchmarks.

**Strengths:**

This paper provides a meaningful conceptual clarification and a practical normalization mechanism that helps stabilize a widely used—but often fragile—class of DFL methods. The insight linking stability radius with learning dynamics is both useful and broadly relevant.

**Weaknesses:**

1. I found some notations and definitions are not rigorous in the paper, see Questions.

2. The paper lacks discussion on other perturbed optimizers beyond the MILP case.

**Questions:**

1. The dimension of Eq. (6) does not seem correct. What is the dimension of the regret loss $L^r$?

2. $f$ is piecewise constant and hardly differentiable. Why you can still write $\nabla_\theta f(\theta)$? It does not seem well defined.

---

> ### Author Response · Authors · 2025-11-14
> **Answers to Reviewer WL6A**
>
> We thank the reviewer for the time and effort invested in the evaluation of our work. We propose to answer all questions and issues pointed out in their report.
>
> **Q: The paper lacks discussion on other perturbed optimizers beyond the MILP case**
>
> A: Our goal is to address issues related to solution stability in decision-focused learning. These issues occur specifically when the objective function is linear, leading to invariance, and when the solution set is polytopic, such that an optimal solution located at an extreme point of the polytope has a non-zero stability radius.
>
> Consequently, this work fully applies to MILP models but does not readily extend to more general optimization problems beyond the MILP setting. Indeed, other non-linear convex optimizers do not share the same difficulty in identifying relevant descent direction in DFL learning. That is the case for quadratic optimizers for example. Nonetheless, MILP models remain by far the most prevalent within DFL pipelines, making our contribution significant in that regard.
>
> **Q: The dimension of Eq. (6) does not seem correct. What is the dimension of the regret loss ?**
>
> A: We thank the reviewer for highlighting the lack of clarity in the definition of the regret loss. The defined regret loss is of dimension 1, in other words it is a scalar value as expected of a standard loss function. Indeed, it corresponds to the difference of two vector products, each between a cost vector $\theta$ and a decision vector $f(\theta)$  both belonging to set $\mathbb{R}^n$, such that $\theta^\intercal f(\theta) \in \mathbb{R}$. This has been clarified in the revised manuscript, along with a reframing of section 3.1 that is now more centered around the DFL pipeline associated to a regret loss.
>
> **Q: $f$ is piecewise constant and hardly differentiable. Why you can still write $\nabla_{\theta} f(\theta)$ ? It does not seem well defined.**
>
> A: Indeed, we apologize for the confusion regarding the differentiability of the optimization mapping $f$. In most cases, $f$ is piecewise constant, which implies that its gradient is zero almost everywhere. Since this gradient has limited relevance in an optimization context, DFL techniques are employed to identify meaningful descent directions for minimizing the loss function. The article has been revised to better reflect this particular characteristic.

---

> ### Author Response · Authors · 2025-11-25
> **Official Comment by Authors**
>
> We sincerely appreciate the reviewers' valuable feedback. We have carefully addressed the concerns raised and hope that our responses have provided sufficient clarification. If you have any remaining questions that require further discussion, we are happy to discuss.
>
> Could you please let us know whether our rebuttal resolves your concerns? We look forward to your further comments and feedback.

---

### Author Response · Authors · 2025-12-02
**Official comment by Autors**

We summarize below our responses to Reviewers and the key points discussed during the rebuttal phase.

All Reviewers acknowledged the relevance of our conceptual clarification regarding stability issues common to decision-focused learning (DFL) approaches.

**(i)** Minor corrections were made, and additional examples were added to better illustrate the theoretical results on solution stability.

**(ii)** Two Reviewers noted a lack of rigor in our introduction of the optimization mapping $f$. We initially described it as “hardly differentiable,” whereas it is piecewise constant. **This has been corrected**: the revised manuscript clarifies that $f$ is differentiable, but its gradient is of limited practical use due to its piecewise constant nature.

**(iii)** Reviewers 5AaY and xxyc misunderstood the nature and purpose of our regularization strategy. Reviewer 5AaY believed it addressed a different problem than solution stability; this was clarified during rebuttal, and the manuscript was improved accordingly. **Reviewer 5AaY updated his score once this misunderstanding was clarified, as stated in the rebuttal discussion.** Reviewer xxyc suggested our approach only aids hyperparameter tuning, despite our explicit statement that it prevents learning degeneration during training. To prevent similar confusion in the future, we have expanded the explanation on the nature and purpose of our regularization strategy and added a practical example in the revised version.

**(iv)** Reviewers 5AaY and xxyc raised concerns about the limited performance gains in numerical experiments. We explained that instability in DFL training is unpredictable and cannot be reliably reproduced without unrealistic conditions. Using state-of-the-art datasets (Mandi et al., 2024), we showed that regularization offers little benefit when instability is absent but significantly improves performance when instability occurs. **Reviewer 5AaY agreed with this observation.**

**(v)** Finally, regarding Reviewer xxyc’s critique, we note that the review does not provide detailed reasoning for the assigned score or specific feedback on the paper’s contributions. The main critique is that our approach “makes sense” but allegedly “does not add a great deal of information compared to what is already known.” **This statement is not supported by references or evidence and appears to rely on assertion rather than substantiated argument.** We kindly ask the Area Chair to consider this when evaluating the paper and to weigh the review’s limited engagement against the other, more detailed reviews.

---

### Meta-Review · Area_Chair_Vhm1 · 2026-01-04

**Summary:**

This paper attempts to formalize the link between perturbation-based methods and solution stability in Decision-Focused Learning (DFL). The authors propose a cost regularization method to manage the stability radius and prevent learning degeneration. The paper provides a rigorous analysis of the relationship between perturbation intensity and solution stability, and validates the proposed regularization method through experiments. A major concern, not fully addressed, is the limited novelty of the paper's core insights. While the formalization is new, the underlying observation that the scale of perturbation affects stability may already be informally understood within the DFL community. The theoretical analysis, though rigorous, does not appear to lead to a breakthrough or a significantly new understanding of the problem. More importantly, the empirical validation is unconvincing. As pointed out by Reviewer xxyc, the proposed regularization strategies do not yield consistent or large improvements. While the authors were successful in convincing one reviewer (5AaY) to raise their score from a 4 to a 6 by clarifying a misunderstanding, this does not override the more substantial concerns.

In summary, while the paper is well-written and tackles a relevant theoretical question, its contribution is ultimately too incremental and its practical benefits are not clearly demonstrated. The weak empirical support is a critical failure that cannot be overlooked. However, a careful evaluation of the reviews and the author's rebuttal reveals that the paper's contributions are not substantial enough to warrant publication at ICLR.

**Reviewer Concerns:**

* A major concern, raised by Reviewer xxyc and not fully addressed, is the limited novelty of the paper's core insights. While the formalization is new, the underlying observation that the scale of perturbation affects stability may already be informally understood within the DFL community. The theoretical analysis, though rigorous, does not appear to lead to a breakthrough or a significantly new understanding of the problem.

* More importantly, the empirical validation is unconvincing. As pointed out by Reviewer xxyc, the proposed regularization strategies do not yield consistent or large improvements. The authors argue that the method is most beneficial in cases of training instability, which are difficult to reproduce. The lack of strong, positive empirical evidence has not been addressed.

**Reviewer Scores:**

* Reviewer WL6A and 8gqm: Their lack of engagement suggests their initial concerns were not resolved. Their scores would likely remain a 4 and a 6, respectively.

* Reviewer xxyc: This reviewer would maintain their score of 2.

* Reviewer 5AaY: This reviewer has already raised their score to 6.

A full discussion period would likely have highlighted the deep disagreement regarding the paper's contribution and the weakness of the empirical results, making a consensus for acceptance unlikely.

---

### Decision · Program_Chairs · 2026-01-26

Reject